

# Use an idealized protocol to assess the nesting procedure in regional climate modelling

Shan Li[1,2], Laurent Li[1], Hervé Le Treut[2]

[1]Laboratoire de Météorologie Dynamique (LMD), Sorbonne Université, CNRS, École Normale Supérieure, École Polytechnique, 4 Place Jussieu, 75252 Paris Cedex 05, France

[2]Institut Pierre-Simon Laplace, Sorbonne Université, 4 Place Jussieu, 75252 Paris, France

*Correspondence to*: Shan Li (Shan.Li@lmd.jussieu.fr)

**Abstract.** Newtonian relaxation allowing RCM to follow GCM is a widely used technique for climate downscaling and regional weather forecasting. A thorough assessment on effects of the relaxation procedure in an idealized framework is presented in this paper for both synoptic variability and long-term mean climate. LMDz is a global atmospheric general circulation model that can be configured as a regional model if the outside domain of the focused region is applied with a relaxation. It thus plays the role of both GCM and RCM in this paper. Same physical parameterization and identical dynamical configuration are kept to ensure a rigorous comparison between the two models. The experimental set-up that can be referred to as "Master (GCM) versus Slave (RCM)" considers the GCM as the reference to assess the behavior of RCM. A further simulation with RCM in a higher resolution configuration allows isolating the effect of relaxation procedure from that of mesh refinement. In terms of mean climate in GCM and RCM, there are noticeable differences, not only in the border areas, but also within the domain. In terms of synoptic variability, there is a general spatial resemblance and temporal concomitance between the two models. But there is a dependence on variables, seasons, spatiotemporal scales and spatial mode of atmospheric circulation. Winter/Summer has the most/least resemblance between the RCM and the GCM. A better similarity is noticed when atmospheric circulations manifested on large scales. No-correlation cases can be remarked when the dominant circulation of the region is at a small scale.

**Keywords.** RCM/GCM, climate downscaling, Master/Slave protocol, Newtonian relaxation, driving forcing, internal variability.

## 1 Introduction

General Circulation Models (GCMs) are the most advanced tools available to study climate variation at global scale. But they generally have a too-coarse spatial resolution (about hundreds of kilometers) to investigate regional climate. Climate downscaling is thus a necessary step for all issues on regional impacts of global climate change. Climate downscaling in the so-called dynamic approach is generally carried out with a physically-based regional climate model (RCM).

RCMs play an essential role in understanding climate variability and impact of climate change behaviors at regional and local scales (Laprise, 2008; Rummukanien, 2010; Giorgi, 2015). Due to the fact that an RCM is formulated, in majority of the cases, over a limited area, one can go to high spatial resolution with limited computing resources. It can be driven by various driving models such as the reanalysis, GCMs and other RCMs. It is cheaper than a GCM at the same resolution. An RCM generally provides an improved climate simulation,



especially with respect to statistical properties of extremes, such as cyclones, intense precipitation and strong winds (Giorgi and Mearns, 1991).

Meanwhile, RCM is far from being a perfect solution for all needs of climate downscaling. RCMs bring added-values with respect to GCM or reanalyzes, but they can also take the drawbacks or retained-value. Many challenges still require attention and efforts. First of all, RCM is of constrained modeling with nudging applied at the lateral boundaries. Nudging is a simple operation that can be realized by adding a "Newtonian relaxation" in the dynamical equations governing the evolution of wind, temperature and humidity (Drobinski, 2015). The Newtonian relaxation added into the prognostic equations of the model is therefore not a physical term, it can introduce problems of boundary inconsistency in the model, but it is a simple and efficient way to drive the RCM. It allows us to use outputs of general circulation model (GCM) of coarse resolution as lateral boundary conditions (LBC) to run high-resolution regional simulations that evolve over time with RCM. A second caveat in RCM is the lack of interactive exchanges between RCM and its driving GCM, since the one-way nesting (OWN) with a unidirectional nudging is the standard methodology to pilot RCM through the outputs of GCM.

In order to understand the characteristic of the RCM, it is necessary to separate the various influencing factors to study downscaling ability of nested regional climate models through the separation of mesh refinement, the influence of downscaling methodology and interaction between different scales.

The present work is devoted to dealing with the problem of the Newtonian relaxation. To isolate its effect, and to understand its role in regional climate simulation, it is necessary to use an idealized framework excluding differences in space resolution and in model physics. The work presented in this paper explores through a methodological study to evaluate essentially the effect of relaxation technique to the regional climate representation. Spatial resemblance of intra-seasonal variability between RCM and GCM is exclusively focused to asses the effect of Newtonian relaxation to RCM.

The study is organized as follows. In section 2, we present the experimental design and data analyzed. Assessment methodology will be introduced in section 3. Section 4 evaluate the effect of Newtonian relaxation to the RCM in 5 subparts. Section 4.1 show the comparison on the seasonal mean between the RCM and the GCM. The spatial resemblance of the circulation within the domain will show in section 4.2 EOF analysis and weather regimes analysis will assess in section 4.3 to decompose regional modes. Section 4.4 analyze the relationship between the external forcing coming from the GCM and the resemblance between the RCM and the GCM. And the effect of the mesh refinement will be presented in section 4.5 before the conclusion.

## 2 Model and Experimental design

The LMDz4 model (Hourdin et al., 2006; Li, 1999) is the atmospheric component of the coupled model IPSL-CM4 (Marti et al., 2005) of the Pierre Simon Laplace Institute (IPSL). IPSL model was largely used to perform climate simulations contributing to the IPCC reports (IPCC, 2007, 2013). It can be operated as a GCM and also as a RCM according to its configurations. In this study, although GCM and RCM are identical, their geographical coverage differs: the GCM covers the entire globe, while the RCM is effective only in the regional





domain considered. Our protocol of simulations can be qualified as "Master versus Slave", since both GCM and RCM are identical, but they are operated in a different way: GCM is entirely autonomous but RCM is driven at boundaries by outputs from GCM.

5    RCM in this study is configured as a large domain extending from the equator to Greenland (latitude: between 2.4° south and 82.4° north) and from the middle of the North Atlantic Ocean to the Caucasus (longitude: between 40.4° west and 42.4° east). This domain covers regions with varied and complex characteristics, such as the North Atlantic, Europe, the Mediterranean and North Africa. It therefore includes several sub-regions commonly used in CORDEX studies (Europe, Mediterranean, Africa, 10   http://cordex.org/community/domains.html). Jones et al., (1995) had shown a domain size of RCM should be large enough to allow the full development of fine scales but small enough to maintain suitable control by the driving lateral boundary conditions (LBC). The choice of the domain size is still a question to study. Our domain presents strong internal variability, especially in mid-latitude regions of the Northern Hemisphere. The internal variability is stronger in summer than in winter (Lucas-Picher et al., 2008; Caya and Biner, 2004; Giorgi and Bi, 15   2000).

The protocol "Master versus Slave" used for this study has a certain resemblance to that of "Big brother versus Little brother" (BBE) proposed by Laprise et al., (2002). BBE protocol consists of performing firstly a GCM with a very high resolution, the same as that in the RCM. Horizontal resolution is then degraded to that of a 20   normal GCM. Degraded information is ultimately used to drive the RCM. Thus the climate simulated by GCM with enhanced spatial resolution (called "Big Brother") plays the role of reference to assess the climate simulated by RCM (called "Little Brother"). Difference between "Big brother" and "Little Brother" obviously reveals the upmost theoretical performance of the RCM. This protocol is particularly interesting for cases where there is no reliable high resolution dataset to evaluate the RCM. Their domain of interest is on the East Coast area of North 25   America, and their model used is the CRCM (Canadian Regional Climate Model) (Caya and Laprise, 1999). Although their simulations cover only a month (February 1993), they were able to conclude that the one-way nesting (Davies and Turner, 1977) applied to the RCM did a good performance in climate downscaling from large scales to regional scales.

30   The common point with the present study is the concept of prefect model which makes it possible to assess the downscaling approach and the operational procedure by getting rid of physical imperfections of the climate model used. GCM is considered as a perfect model and served as the reference for RCM.

Our hypothesis in designing "Master versus Slave" simulations is that there are conceptually two factors 35   affecting the climate downscaling: the general driving methodology of RCM by GCM and the mesh refinement in RCM. To eliminate the effect of RCM resolution, our "Master versus Slave" protocol keeps purposely the two models identical and with the same resolution, about 300 km, a grid of 3.75° in longitude and 2.5° in latitude. In the vertical, there are also 19 identical levels for the two models.





The particular design of our simulation allows us to have a rigorous comparison between the RCM and the GCM, since they are actually identical in terms of physics and spatial resolution. This configuration will be hereafter noted as "DS-300-to-300", standing for downscaling from 300 km to 300 km. A comparison between the two models reveals purely the impact of Newtonian relaxation procedure used for the RCM.

To evaluate the only effect of RCM grid refinement, we actually performed a second simulation, just as "DS-300-to-300", in our protocol "Master versus Slave", but the RCM (Slave) has a higher spatial resolution (100 km, against 300 km in the initial configuration). This additional experiment will be noted hereafter as "DS-300-to-100", standing for downscaling from a model of 300 km as spatial resolution to another model of 100 km. A

relevant comparison between the two experiments can reveal the effect of mesh refinement in RCM, the effect of Newtonian relaxation being eliminated. All simulations of two experiments have a 360-day calendar (30 days for every month). To ensure a good statistical significance, they have all a long duration of simulation of more than 80 years.

The relaxation time scale (τ) represents the nudging strength. When it is smaller than 6h, the nudging is considered strong (Salameh et al., 2010). In this study, all variables (Winds, Humidity, Temperature) U, V, Q and T are strongly nudged since τ is set at 90 minutes. Both GCM and RCM share the same low boundary conditions with climatological sea-surface temperature and sea ice concentration obtained from 1971 to 2000. They also share the same climatological values for greenhouse gases and aerosols over the period 1971 to 2000.

### 3 Assessment methodology

Numerical climate model simulations are affected by various uncertainty due to internal variability, and sensitivity to initial conditions and to boundary conditions (Giorgi, 2006; Stainforth et al., 2005; Murphy et al., 2004). The evaluation of the RCM is often based on the average state to the observations or the GCM. There are

few studies focused on the synoptic scale.

The RCM assessment should be based not only on regional mean climate reproduction, but also on climate variability. We need even to assess the synoptic sequences if the RCM is destined for weather forecast. Intuitively, we can imagine that the reproduction of regional climate depends on two factors, the external forcing

from GCM (reproducible component depend on boundary forcing of GCM) and the internal dynamics (non reproducible) that develops independently in GCM and RCM. Even in a very restrictive framework, the internal dynamics developed within the region can be quite spontaneous (Separovic et al., 2015, 2008, Christensen et al., 2001). Whatever is the climate downscaling protocol, the internal dynamics can occur and makes the RCM to drift significantly from the GCM. Generally speaking, the internal dynamics can come from a better resolution in

the RCM, an advanced physics and the climate downscaling protocol itself. In the past, the protocol issue has never been properly evaluated, since it could not be easily isolated. This is just the focus of our current study. Our basic working hypothesis is that large-scale information of regional climate in the RCM should be consistent with that of the GCM because the RCM is under the constraints of the GCM. At the same time, we recognize that, within the domain, regional climate dynamics can also be generated by internal processes. We

thus expect, when the GCM exerts a dominant constraint on the regional climate, to obtain a good resemblance





between the RCM and the GCM. On the other hand, when the large-scale circulation is weak and the internal dynamics is strong, the RCM is expected to diverge substantially from the GCM.

### 3.1 Data filtered to retain synoptic-scale variability

Daily data of the geopotential at 500 hPa (Z500) and the temperature at 2 meters (T2M) are used to assess the reproduction of the large-scale atmospheric circulation. Separovic et al. (2008, 2015) have shown that the relaxation procedure impacts firstly synoptic (intra-seasonal) scale, an essential element of the atmospheric general circulation (Christensen et al. 2001; Separovic et al., 2015, 2008). By the way, synoptic (intra-seasonal) situation is a very important criterion to represent the internal variability (Separovic et al., 2008, 2015;

Alexandru et al., 2007; Christensen et al., 2001; Jones et al., 1995). The comparison on the reproduction of intra-seasonal between RCM and GCM is chosen to characterize the resemblance between two models who reveals the influence of Newtonian relaxation to regional climate. In other words, it can show the dependence between RCM and GCM following the relaxation procedure. In order to isolate the intra-seasonal variability, daily data have been linearly decomposed into four components, namely the mean state, the inter-annual variation, the

seasonal cycle, and finally the intra-seasonal variation (including synoptic and higher-frequency variability).

### 3.2 Spatio-temporal resemblance in different modes and regimes

The spatial correlation is performed on the filtered data between RCM and GCM to assess the resemblance of the two models. Temporal evolution of this spatial correlation coefficient is examined to trace the daily variation

of the spatial similarity between the two models. EOF analysis and weather regimes analysis are also used in this study to distinguish different situations in order to understand impacts of different scales and modes on the resemblance between RCM and GCM.

## 4 Results

### 4.1 Seasonal Mean Climate

Figure 1 displays the difference in seasonal average on air temperature at 2 meters between RCM and GCM. RCM reproduces well the climate simulated by GCM. This is generally true over all the four seasons (DJF a., MAM b., JJA c., and SON d, Fig. 1). However, there is a significant cooling of more than 1 °C at the eastern boundary for all seasons (Fig. 1). Furthermore, differences are also observed within the domain. The differences

are quite pronounced during DJF and JJA, with a warming of the order of 0.3 °C in sub-Saharan Africa and over the Atlantic Ocean, and a cooling of about 0.6 °C for the summer (JJA) in Eastern Europe. The differences are the strongest in summer. That means the seasonal characteristics could impact the reproduction of the regional climate. The verification of the seasonal mean between RCM and GCM reinforces the initial hypothesis that the downscaling procedure through nudging guarantees a reproduction of large-scale atmospheric circulation

simulated in GCM.

### 4.2 Resemblance of atmospheric circulation within the domain for different seasons

The protocol "Master versus Slave" provides an idealized (but ideal) framework to evaluate effect of the downscaling procedure. Recall that in this study, RCM and GCM have the same physical and dynamic

configurations, apart the relaxation procedure applied in RCM. In the previous section, the comparison between





RCM and GCM shows a significant difference not only at the boundaries but also within the domain. The cause is probably the manifestation of certain autonomy of the internal dynamics in RCM.

The evolution over time will be studied in this section. The filtered daily data are used to keep only the day-to-day variability. That means, for every day, we can compare a physical field between RCM and GCM. The resemblance between RCM and GCM can be measured by the spatial correlation coefficient of the fields between the two simulations.

The objective of this section is to investigate if the relaxation procedure modifies the day-to-day variation within the domain. We examine the spatial resemblance between RCM and GCM for the four seasons. The geopotential at 500 hPa and the temperature at 2 meters are selected to show the near-surface situation and describe the atmospheric circulation at higher altitudes.

The ensemble of spatial correlation coefficients forms a complex distribution that can be represented in a box plot graph. Results are shown either for whole data or for the four seasons separately. The averages in the form of a small dot in the box plots are all below the medians (Fig. 2). This relationship between the mean and the median reveals a biased distribution and the presence of a small number of very small values. At the same time, the spatial correlation coefficients have also a tendency toward high correlation. In fact, a Fisher z-transformation would give approximately a normal distribution, since fields from RCM and GCM can be considered as identically distributed and independent.

The box plots for T2M (Fig. 2) and Z500 (Fig. 3) show all an obvious seasonal variation. A high spatial correlation with a small dispersion (interquartile gap) is found in winter. That is, winter represents a better spatial resemblance between RCM and GCM. Summer shows the lowest spatial resemblance between the two models for both T2M and Z500.

The largest correlation coefficient from T2M is 0.98 (maximum) in winter, 0.90 in summer and 0.96 in spring and autumn. A difference between two high correlations (close to 1) is not easy to detect on the box plots because the values are very close. The 1% percentile values for all four seasons show a peculiarity of very low resemblance in summer with a correlation coefficient of 0.10.

The box size (interquartile gap) and the gap of outliers are two criteria to represent the dispersion of spatial correlation coefficients between RCM and GCM. Seasonal characteristics are clearly shown on the box plot. There is a larger dispersion in summer than in winter (Fig. 2). Temperature at four other pressure levels (1000 hPa, 850 hPa, 500 hPa and 300 hPa) are also analyzed. The results (not shown) largely confirm those found on the temperature at 2 meters.

Figure 3 summarizes the statistics for correlation coefficients for the 500 hPa geopotential height. For whole data and the four seasons, a good correlation is noticed with an average exceeding 0.80 and a median exceeding 0.90. The 99[th] percentiles all exceed 0.99. The spatial correlation coefficients for Z500 show the same seasonal





variation as for T2M. RCM has the best reproduction of GCM in winter and the worst in summer. However, the spatial correlation coefficients for Z500 (Fig. 3) are higher than for T2M (Fig. 2), with also a smaller dispersion. This is shown for all statistics (mean, median, quartiles). This phenomenon shows that there is a better reproduction at altitudes than near the surface. The reproduction in RCM is more impacted at the surface than at
altitudes.

### 4.3 Main modes of regional variability

The spatial correlation coefficient analysis in the previous section shows there is a better resemblance between RCM and GCM in winter. The particularity of winter should be related to the strong variance in its general
circulation (not shown). The study domain is dominated by the NAO mode, but other modes of regional variability also exist. The different modes should have a distinct reaction to the relaxation operation imposed to RCM. We perform EOF analysis (Fig. 4, 5) and weather regime (Fig. 7, 8) analysis to stratify the correlation coefficients. The results confirm the initial hypothesis: the two models have non-identical structures, but very close.

The analysis is applied to the filtered daily data of the geopotential at 500 hPa, for the day-to-day variability of the atmospheric circulation. The EOF analysis gives in descending order of interest the spatio-temporal patterns, which explain the most variability and leave the noise in the EOF of high order. In order to compare the time series (PC: principal component), it is necessary to have a series of common spatial structures for RCM and
GCM. The combined EOF is used to characterize the ten first spatial structures (over 92% contribution, Fig. 4). The analysis is done separately over the four seasons. However, only winter will be shown because of the strongest resemblance between the two models in winter. Similar results are noticed in three other seasons (not shown).

Figure 4 shows the decomposed structures for winter. The spatial structures are presented from large scales to small ones, with a decreasing order in the explained variance. The first three EOFs show large-scale structures, which have a contribution of 64.97%. The first EOF shows essentially a north-south bipolar structure between the Greenland Sea and the Mediterranean Sea. It represents the North Atlantic Oscillation, the most important mode of variability in this region. The second EOF also represents a bipolar structure (Fig. 4), but with contrasts
between the East (Central Europe) and the West (Middle North Atlantic). The third EOF shows a remarkable oval structure (Fig. 4), centered in the North Sea with an extension from the middle of the Atlantic to the Caucasus. At the same time, there is a weak expression with opposite sign towards Greenland and the Red Sea. It seems that this structure is in very weak relation with the outside, because it has practically no expression in border zones.


The fourth and fifth EOFs both represent a structure like horse-saddle (Fig. 4). Their contribution in variance also remains very close, and around 7.5% for both. They represent a structure that propagates: a movement in counterclockwise rotation is visible between these two structures. The sixth EOF is an oval structure stretched from Greenland to the Barents Sea with a center on the Norwegian Sea. This structure is encompassed by
opposed values, with a strengthened expression in the middle of the North Atlantic, the Balkan and the Arctic



Ocean. Higher order EOFs (EOF7, EOF8, EOF9 and EOF10) show smaller scales structures with a wave number around 2.0 (Fig. 4).

With fixed common spatial structures, it is now useful to compare the corresponding time series between RCM and GCM. Our purpose is to check if some modes promote/disadvantage a good/bad temporal concomitance between RCM and GCM. Remember that RCM is a constrained model, with control, at the outside of the domain, from GCM, through a relaxation operation. It is expected that different structures have their own behaviors in response to constraints from the boundaries. In other words, the influence of external forcing from the outside of the domain should be different for different spatial structures. We perform then a correlation
calculation between the two corresponding temporal series for each EOF structure to show how their similarity varies for different dominant modes. The reproduction in RCM of the temporal variation of GCM is represented by a correlation coefficient (Fig. 5). A low temporal correlation coefficient reflects non-concomitant variations between the two models. In other words, a low temporal correlation coefficient means that the two models do not vary at the same time in the same mode.

For the case of Z500 in winter, the two models are close to each other for all the ten first EOFs. The correlation coefficients are all greater than 0.84. A very strong resemblance (correlation coefficient greater than 0.95) is found on the first five EOFs (Fig. 5). On the other hand, EOF3 has the lowest correlation coefficient (0.93) among the first five EOFs, but it is still greater than those for smaller scale modes (from EOF6 to EOF10) which
have a coefficient around 0.90 (Fig. 5). For the first two EOFs which contribute nearly half of the variance to the physical field, RCM and GCM are extremely close to each other with a high correlation coefficient larger than 0.97 (Fig. 5). The trend line in Figure 5 clearly shows that the concomitance between the two models decreases from large scales to small scales.

It is clear that the effect of the relaxation operation is dependent on spatial scales. The relaxation procedure operated in this study creates a favorable situation for RCM to have greater freedom at small scales than at large scales (Fig. 5).

Figures 4 and 5 reveal that the control from GCM to RCM depends not only on spatial scales, but also on spatial
modes. Some modes show a weaker concomitance between the two models such as EOF3, EOF6 and EOF9. They have all an oval structure around 60° N (Fig. 4) and poorly connected to the boundaries. This is especially true for EOF3 in which the isolated oval structure presents a large geographical extension covering Europe and the North Atlantic. The oval structure noticed in these three EOFs makes RCM easier to have greater freedom to manifest its own behaviors, and the temporal evolution between the two models is less reproducible.


The EOF analysis confirms that the downscaling procedure with the relaxation operation diverts the spatio-temporal variability of RCM from GCM, although their divergence remains weak.

The relaxation procedure operated in this study ensures a good simultaneity between the two models at large
scales (Figure 4, 5, 6). RCM shows more freedom at small spatial scales. This can be furthermore demonstrated





with reconstructed fields from different scales. We will consider three cases with physical fields reconstructed from the first 10 EOFs.

The first ten EOFs explain about 92% of the total variance. The first five EOFs explain 79%, and they are
mainly from variations of large scales. The last five EOFs from EOF6 to EOF10 explain 13% of the total variance, mainly for variations of small scales. With such reconstructed fields, we perform again the spatial correlation coefficients between RCM and GCM. Their distributions in the form of box plots are displayed in Figure 6. Large scales (first five EOFs) show a greater resemblance and smaller dispersion, compared to small scales (between EOF6 and EOF10). It is consistent with the result presented previously, namely, the spatial
resemblance and the temporal reproduction between RCM and GCM are generally dominated by large-scale atmospheric circulations from GCM. At the same time, RCM does show freedom by simulating small-scale circulations which are not necessarily controlled by GCM.

In mid and high latitudes, quasi-stationary states of atmospheric circulations are recurrent and can be easily
recognized at synoptic scale. They are often referred to as weather regimes or circulation regimes. In the geographic sector Europe / North Atlantic, four weather regimes are generally recognized. They are: NAO+ (zonal), NAO-, blocking and Atlantic Ridge. Since different circulation regimes are discriminable for the regional atmospheric circulation and for the regional weather, for example, the zonal regime is linked to winter storms and the blocking regime is associated to cold weather, we can imagine that the resemblance between
RCM and GCM may be very dependent on the regional weather regimes. We believe that the stratification of results into different regimes can provide relevant explanations on the resemblance between the two models. Our objective is to understand why such conditions or others are favorable / unfavorable to bring RCM closer to GCM.

Figure 7 shows the four weather regimes that we obtained with 500-hPa geopotential height from GCM, by using a classic K-means algorithm. They have a similar distribution of presence, each with a quarter occurrence frequency. The regime Atlantic Ridge (regime 1 with 24.18% occurrence frequency), the NAO- (regime 2 of 24.76%) and the blocking (regime 3 of 24.86%) all have a less important occurrence than the zonal (NAO+, regime 4) with an occurrence frequency of 26.19%. This means that our GCM simulates more winter storms
with a stronger presence of the zonal regime (Fig. 7). The resemblance between RCM and GCM is always evaluated by means of the spatial correlation coefficient. Results are displayed in Fig. 8. We remind that a Fisher transformation is performed to facilitate the differentiation of two high values of correlation.

There is not a big difference on the resemblance between RCM and GCM after the stratification into four
weather regimes. However, a bigger difference between the two models is noticed on the blocking regime and zonal regime. Figure 8 shows clearly that RCM has less resemblance to GCM on the blocking regime than on three other regimes (Fig. 8). This means that RCM has more freedom to reproduce the cold winter simulated in GCM. At the same time, there is a better distribution of correlation coefficient (Fig. 8) on the zonal regime. However, a bigger dispersion of spatial resemblance is also noticed at the zonal regime.






### 4.4 Reproductive fidelity of regional circulation according to the boundary conditions

The influence of external forcing from GCM to the resemblance between the two models will be examined in the following. The intensity of the external forcing is diagnosed by the variance of geopotential height at 500 hPa at the outside of the boundary. The 45° West and 45° East boundaries are both close to the study domain (between

40.4° west and 42.4° east). This choice is based on a comparison and verification among a few positions, such as 45°, 50° and 65°. The same relationship between the external forcing from the GCM and the internal resemblance between the two models is found with different positions to characterize the external forcing.

In Figure 9, we can observe that the intensity of external forcing is different between the western (between 0 and

12000 m$^2$, Fig.9.b) and eastern (between 0 and 60000 m$^2$, Fig. 9.a) boundaries. Figure 9 also shows that the resemblance between the two models increases with the intensification of external forcing. This means that a strong GCM control favors to have a good similarity between RCM and GCM. Figure 9 also reveals that low correlation coefficients (less than 0.5) are associated with a very small variance for both west and east boundaries. However, a weak external forcing does not always imply a bad resemblance between the two models.

Table 1 presents a numerical summary of what shown in Figure 9. First, we can see that the two models are generally very close to each other with a spatial correlation coefficient greater than 0.95 in 4396 days out of 7200 (61.05%). On the other hand, there are only 29 days out of 7200 (0.40%) with a low resemblance characterized by the correlation coefficient smaller than 0.5. Second, the variance of the east boundary is smaller

than that of the west on all classes of similarity. Third, the average of the variances of the boundaries has an obvious relation to the correlation coefficient. That means when the correlation coefficient is low, the variance of the boundary is also small, and a high correlation corresponds to a high variance (Figure 9, Table 1).

The interior of the region is more or less controlled by large-scale circulation coming from the outside of the

domain. The strong external forcing manifested by a high value of variance at the boundary, favors a good reproduction of the RCM towards the GCM. On the other hand, a weak external forcing makes the effect of the internal dynamics more important, which causes a divergence to the two models.

### 4.5 Effect of the mesh refinement

In previous sections, all analyzes are based on the experiment without a finer resolution. The application of mesh refinement at the regional scale is necessary because the coarse resolution of the GCM is not sufficient to correctly simulate the regional climate (Giorgi et al., 1991, 2010; Jacob et al., 2007; Laprise et al., 2008; Castel et al., 2010; Rummukainen, 2010; Richard et al., 2010). By the way, the horizontal resolution is an important issue for regional climate modelling. The framework is therefore completed by a second simulation ("DS-300-

to-100") in which the RCM has an increased spatial resolution (100 km), with all other aspects unchanged. The comparison between the two configurations helps to reveal the impact of mesh refinement in the RCM whose effect is added to that of the nesting procedure (Fig. 10).

The bi-histograms in Figures 10.a and 10.b show the same relationship between the external forcing and the

correlation coefficient between the two models. With mesh refinement, a strong external forcing is always





associated with a high value of spatial correlation coefficient and a very low correlation value is always in situations of weak external forcing. In both protocols, there is a large number of very strong correlation with moderate variances (Fig. 10.a, 10.b). A visual comparison between Figure 10.a and Figure 10.b shows a kernel shift to the left. That means from "DS-300-to-100" to "DS-300-to-300", there is a trend toward lower correlation.

The decrease of correlation (resemblance) following the mesh refinement is obviously noticed on the subtraction of two protocols (Fig. 10.c).

The most significant difference between "DS-300-to-100" and "DS-300-to-300" is found in the range with the external variance less than 20000 m$^2$ and a spatial correlation coefficient exceeding 0.70 (Fig. 10.c). Compared

to "DS-300-to-300", "DS-300-to-100" present a decrease about 40% (-0.08/0.2) of the spatial correlation which exceeds 0.93. At the same time, there is an increase in frequency of occurrence for the range of correlations between 0.70 and 0.93. An obvious increase about 60% (0.03/0.05) is found in the correlation between 0.80 and 0.93, and a smaller increase of 30% between 0.70 and 0.80 (Figure 10.c, Figure 10.a).

To make the bi-histogram more symmetrical, a Fisher transformation is applied for the spatial correlation coefficient and a natural logarithm is used for the variance. A shift to the left of the center of high probability (Fig. 10.d, 10.e) is noticed with a decrease in the upper 50 percentiles and an increase in the lower 50 percentiles (Fig. 10.f). The decrease amplitude is larger than the increase. The comparison between "DS-300-to-100" and "DS-300-to-300" clearly shows that the mesh refinement in the RCM decreases the spatial resemblance between

the RCM and the GCM. Furthermore, the influences from boundary conditions remain unchanged between the two experiments.

**5 Conclusion**

This paper was devoted to the investigation of effects of a largely-used climate downscaling procedure which

uses a Newtonian relaxation in order to drive RCM with outputs from GCM. We designed an idealized framework, called "Master versus Slave" in which GCM and RCM are identical, but GCM is operated autonomously and RCM is relaxed to GCM at boundaries. The fidelity of RCM to an identical GCM is firstly analyzed (experiment "DS-300-to-300"). The GCM was used as the reference to evaluate the RCM. We thoroughly examined the spatial-temporal resemblance between RCM and GCM. We also performed the

analysis with a stratification of regional atmospheric circulation into different modes or regimes which are believed to play a discriminant role in the relation RCM/GCM. Finally, the intensity of external forcing is also revealed to be a determined factor for the resemblance between the two models.

In terms of mean climate, RCM can reproduce main spatial patterns of 2-m surface air temperature and 500-hPa

geopotential height as in GCM. But significant differences do manifest, especially at the boundaries, due to the inevitable conflict between imposed external forcing from GCM and internal dynamics in RCM. Beyond the difference found near the boundaries, we also found significant difference for the whole domain. If the former can be simply treated by an exclusion of the boundaries from our analysis, the later may raise serious challenges for climate downscaling.




Beyond the mean climate simulated in RCM, we also examined the synoptic sequences reproduced by RCM and their resemblance to those in GCM. We found that there is a certain dependency on seasons and regional atmospheric circulation modes or regimes. The resemblance between the two models is shown to be strong in winter than in summer, in larger scales than in smaller ones. Furthermore, the blocking regime in the region

seems to have a larger autonomy in RCM. The results are generally in agreement with our expectation, since the reproduction of synoptic sequences is a compromise between the external forcing from GCM and the internal dynamics generated in RCM. The external forcing was thoroughly examined. Strong external forcing promotes a good spatial resemblance and a good temporal reproduction of the RCM towards the GCM. However, the external forcing does not always guarantee to have a good coherence of regional climate simulation between the

two models, because of the impact of relaxation procedure.

The internal atmospheric dynamics come from two sources of variability. On the one hand, there is a relation with the continuity of the movement coming from the outside of the domain and the physic-dynamic law governing the continuity of the general atmospheric circulation. On the other hand, regional climate dynamics

are also generated by local processes within the study domain, independently of what happens outside the region. The internal dynamics has more freedom in refined RCM which is impacted by more detailed surface process. The mesh refinement increases the RCM's autonomy, with less dependence on the GCM. In other words, there is more development of the internal dynamics when the spatial resolution of the RCM is increased. Further results on internal variability and its influences on the reproduction of climate and synoptic sequences in RCM

will be reported in a future work.





*Code availability*. The general description of this model can be found at https://cmc.ipsl.fr/ipsl-climate-models/ipsl-cm4/. The sources of the LMDz4 model are accessible at http://web.lmd.jussieu.fr/trac/browser/LMDZ4. A detailed description of this model can be found in this paper: https://link.springer.com/article/10.1007/s00382-009-0640-6 (Marti et al. 2010). The model code corresponding

to our experiments are accessible at http://www.lmd.jussieu.fr/~lshan/GMD_code/.

*Data availability*. The outputs of different simulations are accessible at http://www.lmd.jussieu.fr/~lshan/GMD_data/. In the directory "oneway_master"(http://www.lmd.jussieu.fr/~lshan/GMD_data/oneway_master/), we can find the outputs of

GCM of our experiment "DS-300-to-300". The outputs of RCM of the experiment "DS-300-to-300" can be found in the directory "oneway_slave" (http://www.lmd.jussieu.fr/~lshan/GMD_data/oneway_slave/). In each subdirectory "reference_frequence12perday_tau90m", we can find the daily data of the geopotential at 500 hPa (Z500) and the temperature at 2 meters (T2M).

*Competing interests*. The authors declare no competing interests

*Author contribution*. All authors designed the experiments. Shan LI analyzed statistically all simulations and prepared the manuscript with contributions from all co-authors. Laurent LI developed the model code and performed the simulations. Hervé Le Treut participated analysis design, study reflection and discussion.

*Acknowledgements*. This paper is about a part of my thesis work at Sorbonne University in the Dynamic Meteorology Laboratory (LMD: Laboratoire de Météorologie Dynamique). Thanks to Sorbonne University, LMD and Institute Pierre Simon Laplace (IPSL). The Institute for Development and Resources in Intensive Scientific Computing (IDRIS) and Mesocentre of IPSL support all simulations and data processing. I would also
like to thank Patricia Cadule for passing comments of this paper, for her scientific suggestion, for the scientific discussion and for her encouragement.

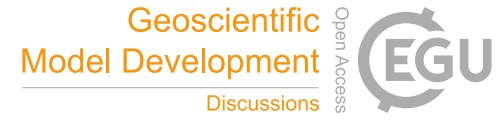



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





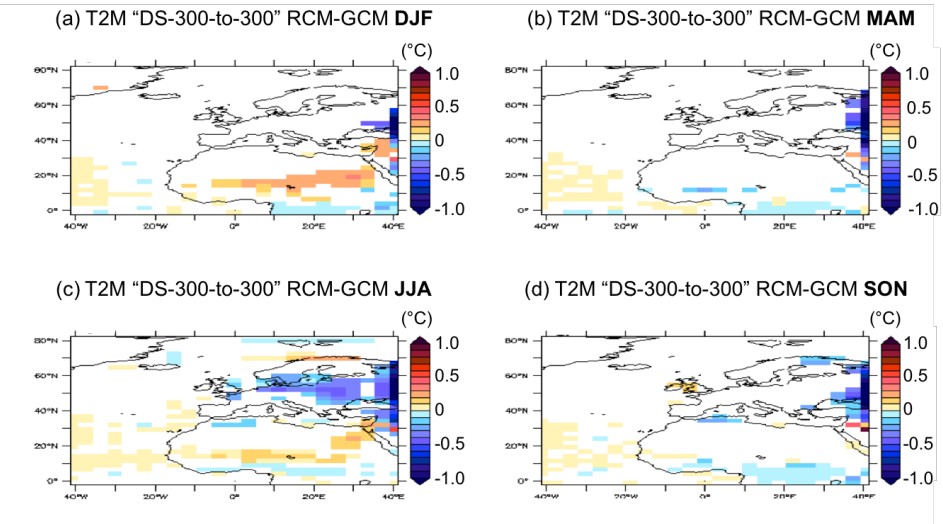

**Figure 1. Differences of seasonal averages between RCM and GCM (served as reference simulation) of surface air temperature at 2 meters. Simulations were conducted within "DS-300-to-300" in which RCM and GCM are identical including the same spatial resolution of 300 km.**

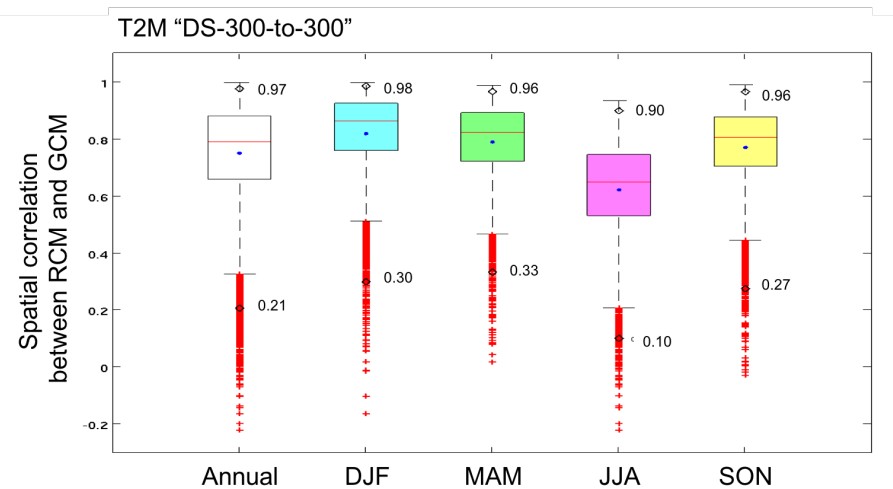

**Figure 2. Box plots showing the distribution of correlation coefficients (between RCM and GCM) for surface air temperature at 2 meters. The calculation is for whole data and 4 separate seasons, respectively. The bleu point is the average. The red line is the median. Red crosses are outliers.**





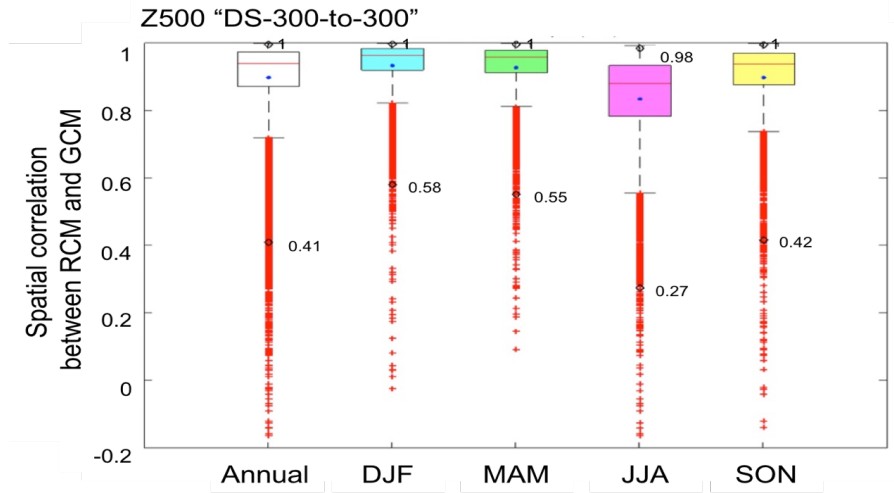

**Figure 3. Same as in Figure 2, but for the geopotential height at 500 hPa.**

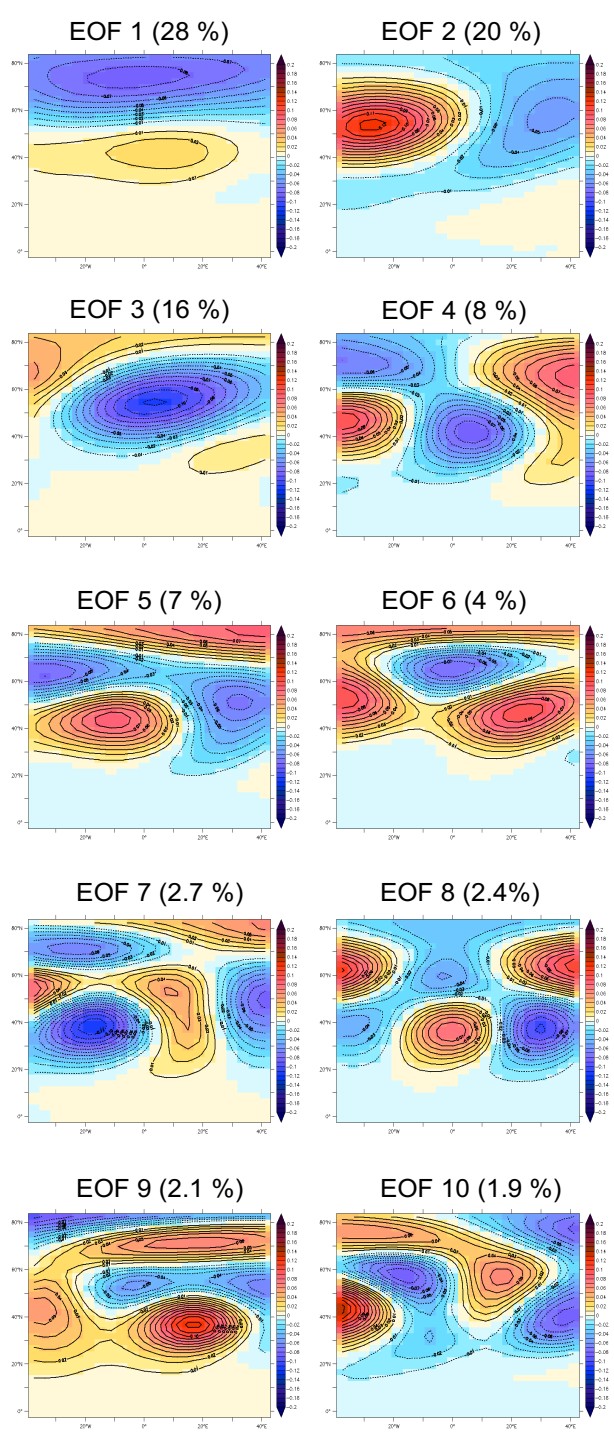

**Figure 4. Spatial patterns of combined EOF from RCM and GCM for winter (DJF) filtered daily Z500 in "DS-300-to-300". Percentages above each chart show the fraction of explained variance.**





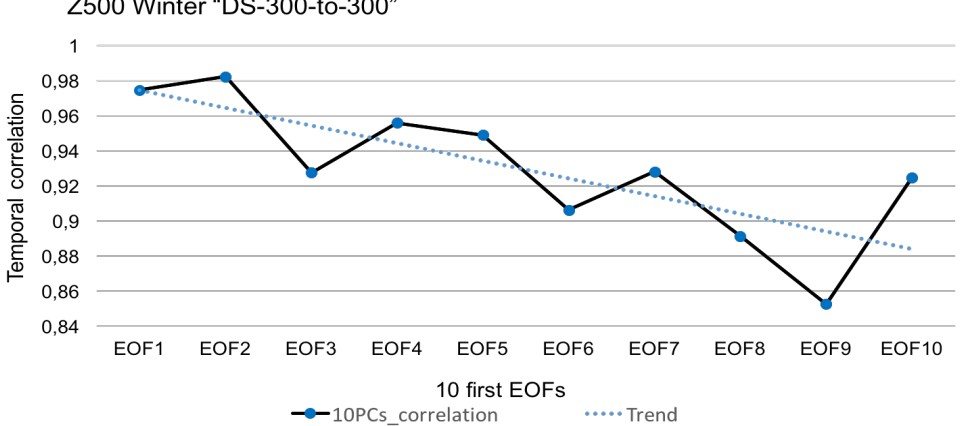

**Figure 5.** Time correlation coefficients between RCM and GCM, for the first 10 EOF structures. The dotted line is a linear regression of the 10 correlation coefficients.

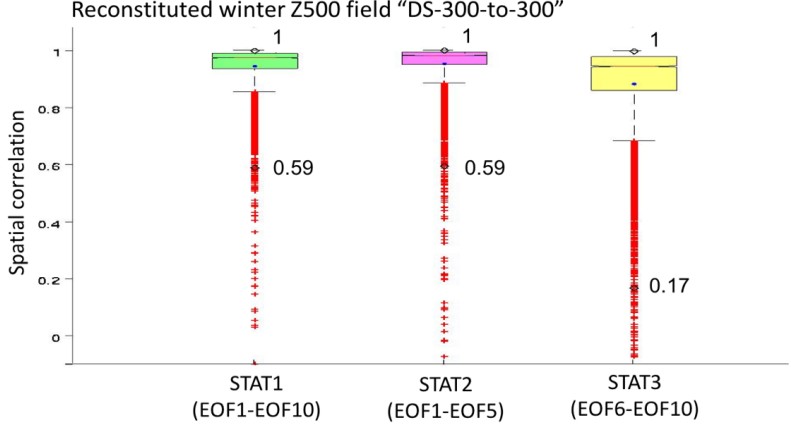

**Figure 6.** Box plots showing the distribution of spatial correlation coefficients between RCM and GCM for reconstituted Z500 fields in winter: STAT 1 represents the 10 first EOFs (92.19%), STAT2 represents the reconstituted Z500 from the 5 first EOFs (79%) and STAT3 represents the reconstituted Z500 from EOF6 to EOF10.



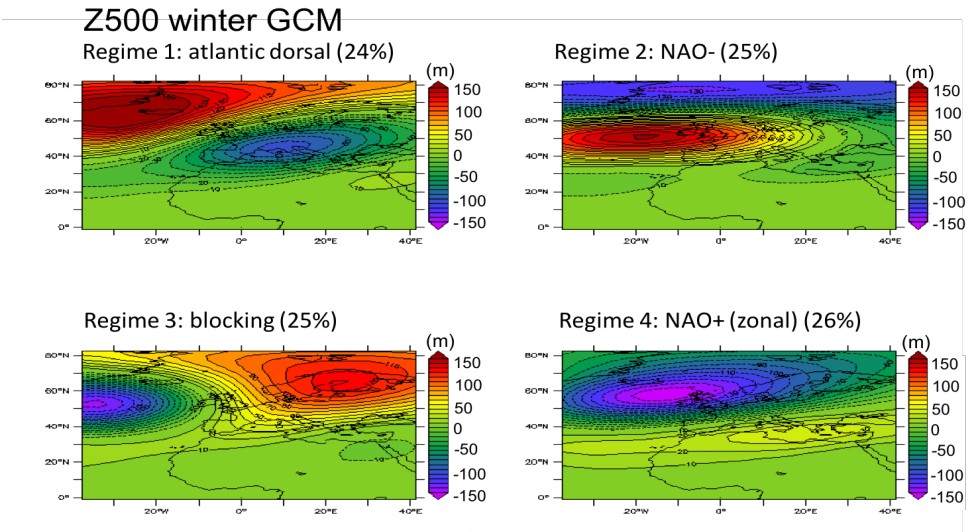

**Figure 7. Four weather regimes of the winter season, calculated with daily geopotential height at 500 hPa in GCM. Regime 1 represents the Atlantic Dorsal with 1783 days (24.18%). Regime 2 is the NAO- with 1741 days (24.76%) over the entire 80-years. The 1790 days of the blocking regime (14.86%) is represented in regime 3. The zonal regime (NAO+) is in regime 4 of 1886 days (26.19%).**

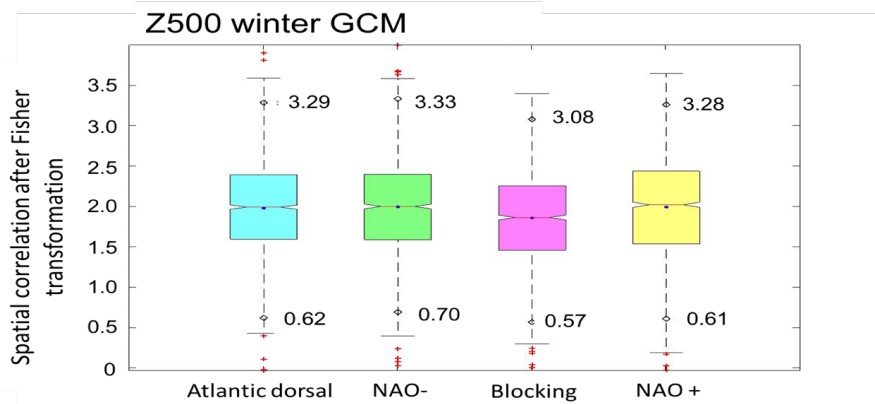

**Figure 8. Box plot of spatial correlation coefficients (after Fisher transformation) between RCM and GCM for winter. They are calculated after stratification on four weather regimes: the Atlantic dorsal, the negative phase of NAO, the blocking regime, and the NAO+ regime.**





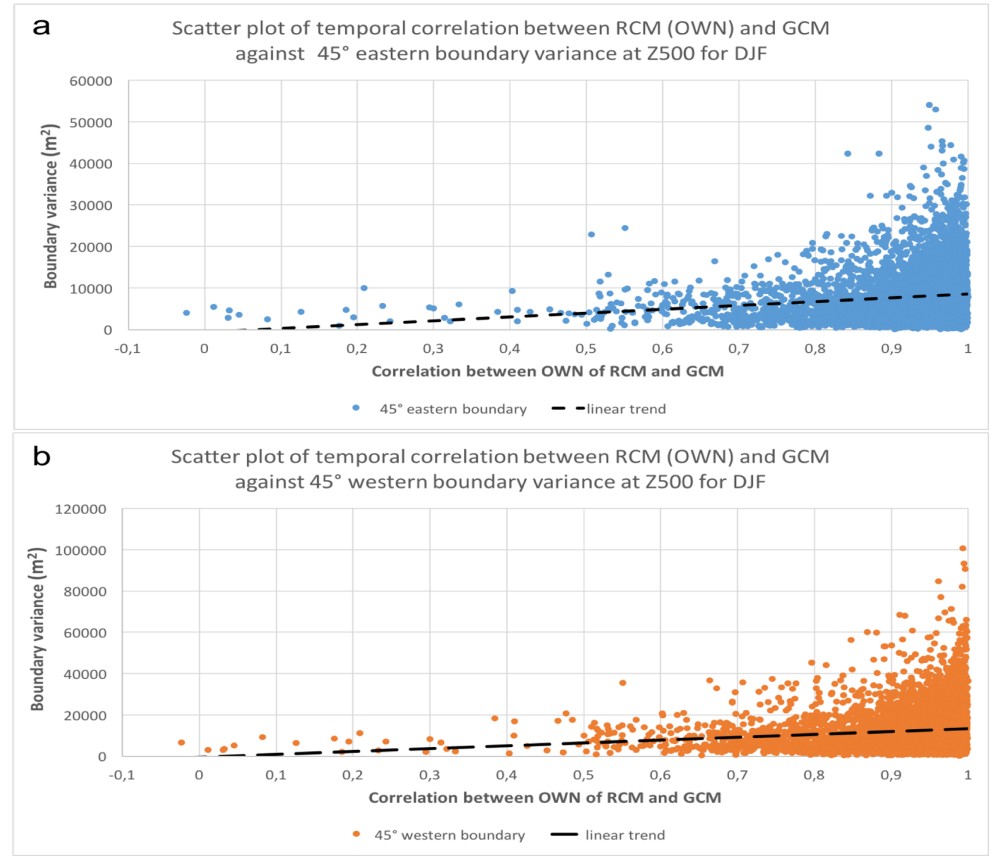

**Figure 9. Scatter plots showing the variance (Y axis) of Z500 (intra-seasonal variability only) at 45° E (a, blue) and 45° W (b, orange). The X axis shows the spatial correlation coefficients between RCM and GCM, calculated for intra-seasonal variability of Z500.**

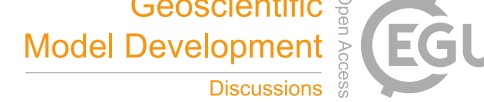



**Table 1. Different classes of correlation coefficients with variances at 45° East and 45° West respectively.**

| Classes of correlation (DJF) | Number of day | Average of variance at 45° W (m$^2$) | Average of variance at 45° E (m$^2$) |
|---|---|---|---|
| -0.3 : 0.5 | 29 | 7436.02 | 4202.60 |
| 0.5 : 0.7 | 181 | 8896.70 | 5305.82 |
| 0.7 : 0.9 | 1184 | 10395.45 | 6611.56 |
| 0.9 :0.95 | 1410 | 11821.19 | 7864.70 |
| 0.95 : 1 | 4396 | 13141.52 | 8396.28 |

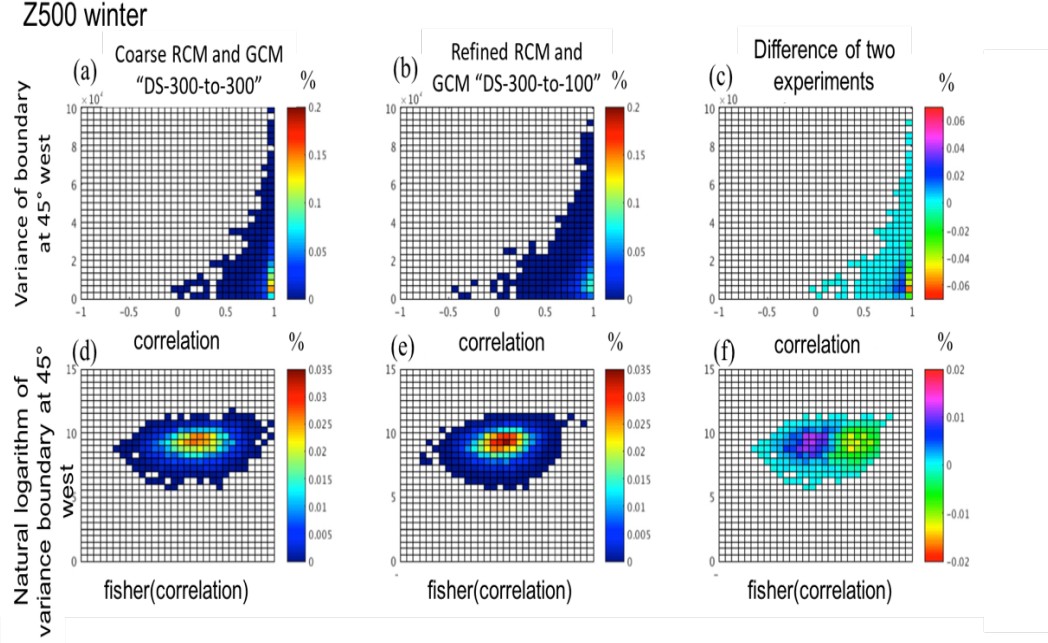

**Figure 10. Standardized bi-histograms (or bivariate probability distribution functions) showing the probability of occurrence as a function of the correlation coefficient and the variance. The spatial correlation coefficient is calculated from the intra-seasonal variability of Z500 between RCM and GCM. The variance is calculated on the intra-seasonal variation of Z500 at 45° W. All results are for DJF. Panels on the left are from the "DS-300-to-300" protocol (a, d), those in the middle are from the "DS-300-to-100" protocol (b, e), and those on the right are the subtraction of the two experiments (c, f). Panels at the top (a, b, c) are from direct calculations. Those at the bottom (d, e, f) undergo a Fisher transformation for the correlation coefficient, and a natural logarithmic transformation for the variance.**

