# Peer review of "Use an idealized protocol to assess the nesting procedure in"

_Geoscientific Model Development, 2018_

## Short Comment (SC1) · 18 Dec 2018

Dear authors,

In my role as Executive editor of GMD, I would like to bring to your attention our Editorial version 1.1:

http://www.geosci-model-dev.net/8/3487/2015/gmd-8-3487-2015.html

This highlights some requirements of papers published in GMD, which is also available on the GMD website in the 'Manuscript Types' section:

http://www.geoscientific-model-development.net/submission/manuscript_types.html

In particular, please note that for your paper, the following requirements have not been

met in the Discussions paper:

- "The main paper must give the model name and version number (or other unique identifier) in the title."

- "If the model development relates to a single model then the model name and the version number must be included in the title of the paper. If the main intention of an article is to make a general (i.e. model independent) statement about the usefulness of a new development, but the usefulness is shown with the help of one specific model, the model name and version number must be stated in the title. The title could have a form such as, "Title outlining amazing generic advance: a case study with Model XXX (version Y)"."

- "All papers must include a section, at the end of the paper, entitled 'Code availability'. Here, either instructions for obtaining the code, or the reasons why the code is not available should be clearly stated. It is preferred for the code to be uploaded as a supplement or to be made available at a data repository with an associated DOI (digital object identifier) for the exact model version described in the paper. Alternatively, for established models, there may be an existing means of accessing the code through a particular system. In this case, there must exist a means of permanently accessing the precise model version described in the paper. In some cases, authors may prefer to put models on their own website, or to act as a point of contact for obtaining the code. Given the impermanence of websites and email addresses, this is not encouraged, and authors should consider improving the availability with a more permanent arrangement. After the paper is accepted the model archive should be updated to include a link to the GMD paper."

Therefore please make the exact version, your article refers to, available via a permanent archive providing a DOI (e.g. Zenodo) and not via a private homepage which

has no guaranty of continuation. Additionally add a reference to the model (including version number) used in the title, e.g.

Use an idealized protocol to assess the nesting procedure in regional climate modelling, a case study with LMDz4

Yours,

Astrid Kerkweg

———————————————

---

## Author Comment (AC1) · 28 Dec 2018

In response to the Executive Editor's comments, we plan to do the following modifications:

1. We accept the suggestion for adding a precision in the title:

"Use an idealized protocol to assess the nesting procedure in regional climate modeling: a case study with LMDZ4"

2. We prepared three Supplements which are downloadable from the LMD's web page:

Supplement I: http://www.lmd.jussieu.fr/∼li/LMDZ4_compilation.docx

Supplement II: http://www.lmd.jussieu.fr/∼li/LMDZ4_code.tar.gz

[Figure]

Supplement III: http://www.lmd.jussieu.fr/∼li/LMDZ4_data.tar.gz

The three files will be uploaded into the GMD's server as Supplements.

3. We will add a new paragraph at the end of Section 2 (Model and Experimental design) in order to make the necessary linkage in Main text to the Supplements:

"For the sake of completeness, and to preserve the traceability of our work, Supplement I (or http://www.lmd.jussieu.fr/∼li/LMDZ4_compilation.docx) provides detailed information and guidance to compile the code and to run simulations presented in this work. Supplement II (or http://www.lmd.jussieu.fr/∼li/LMDZ4_code.tar.gz) provides an archived file containing the code, and Supplement III (or http://www.lmd.jussieu.fr/∼li/LMDZ4_data.tar.gz) (file size 111 Mb) provides configuration files, boundary conditions and job-launching shell scripts".

4. The Section "code availability" becomes:

"The code used in this study is provided in Supplement II (or http://www.lmd.jussieu.fr/∼li/LMDZ4_code.tar.gz) under the License CeCILL"

5. The Section "data availability" becomes:

"Configuration files, boundary conditions and job-launching scripts are provided in Supplement III (or http://www.lmd.jussieu.fr/∼li/LMDZ4_data.tar.gz) under the License CeCILL".

---

## Referee Comment (RC1) · Anonymous Referee #1 · 12 Jan 2019

I have read this paper several times and I cannot determine from the manuscript whether the authors have used Newtonian relaxation around the perimeter of the RCM domain (ie Davies 1976) or in its interior (eg von Storch et al. 2000). Neither the model description nor the experimental design discussion helps clarify this issue. For example, the authors state "RCM is of constrained modeling with nudging applied at the lateral boundaries" (pg 2,lines 6-7), which makes it seem like they are employing the standard Davies approach. However, in the very next sentence the authors state, "Nudging is a simple operation that can be realized by adding a "Newtonian relaxation" in the dynamical equations governing the evolution of wind, temperature and humidity (Drobinski, 2015)." (pg 2, lines 7-8). The reference to Drobinski (2015) would seem to indicate that they are using Newtonian relaxation "everywhere" in the interior of the

RCM domain as this was the central point of discussion in that earlier study.

Clarity on this issue is central to the current study. I would ask that the authors revise their paper indicating more clearly the details of their model configuration and their experimental design. In doing this they also need to indicate, the temporal resolution of the "GCM" driving data and its potential influence on their results; the nature of the blending region (ie the Davies type nudging that is specified around the perimeter of the RCM domain) including its width, profile of strength, and the variables nudge in this region.

Additionally, if the authors are driving the RCM with 6hr GCM forcing (it is not stated), which is most common (ie CMIP5/CORDEX), they need to explain why they feel it is appropriate to nudge the RCM with tau=1.5hr. The instantaneous (6hr) snapshots of GCM winds and temperatures are solutions to both the GCM and RCM governing equations. These GCM forcing fields, however, must be interpolated down to the time step of the RCM. Such interpolated fields "are not" solutions of the GCM/RCM. Forcing the model strongly towards these interpolated fields (ie by using a value of tau well below the 6hr GCM update frequency) would force winds and temperatures into the RCM that are not solutions (ie unbalanced on the largest scales). This would excite gravity waves and produce variance structure, which is presumably captured in the authors analysis. Ideally, the value of tau should be longer than the time that the GCM driving data is updated (eg 6hr).

For these reasons I recommend major revision of this paper prior to me being able to provide a review.

References ————-

Davies, H. C., 1976: A lateral boundary formulation for multi-level prediction models. Quart. J. Roy. Meteor. Soc., 102, 405-418.

von Storch, H., H. Langenberg, and F. Feser, 2000: A spectral nudging technique for

dynamical downscaling purposes. Mon. Wea. Rev., 128, 3664-3673.

---

## Short Comment (SC2) · 14 Jan 2019

The Newtonian relaxation is applied outside the RCM's domain. There is no relaxation at all inside. So our practice corresponds to Davies (1976), but not Von Storch (2000). The confusion was certainly in the reference to Drobinski et al. (2015) who performed relaxation inside RCM. In our original text, we only wanted to mention that our relaxation was applied to the four meteorological variables T, u, v and q, as in Drobinski (2015). To avoid confusion, we will simply delete this reference, and clarify the experimental design.

The temporal resolution of lateral boundary conditions from GCM is every two hours. The buffer zone is indeed the whole globe outside the RCM's domain. In

our "master/slave" configuration, RCM also covers the whole globe. In terms of relaxation strength, we used a binary solution for the relaxation time: 1.5 hour outside the domain and infinity inside (1.0e+25). We didn't employ any transition between the two. In fact, our configuration inherited from a two-way nesting methodology in which we need the two models to be spatially complementary from each other. We now provide our model code and configuration files in the Annex (http://www.lmd.jussieu.fr/~li/LMDZ4_compilation.docx) (please don't copy/paste, but type the link in the browser). All interested readers may totally reproduce our configuration. We will add all these descriptions in the revised manuscript.

We agree with the reasoning of the Referee for the updating frequency of GCM and the relaxation time, although we did not observe any obvious distortions. We recall that our GCM updating frequency is here every two hours, and the relaxation time scale is 1.5 hour for RCM boundaries. The relaxation time is an indication of the e-folding time scale. Relaxed variables can never reach the driving variables. This issue is naturally included in our configuration and in many other RCM practices. It constitutes somehow our investigation objective. This manuscript is a first step to investigate the commonly-used methodology of driving RCM through lateral boundary conditions. Impact of GCM updating frequency and that of the relaxation time scale are planned to be reported in future.

---

## Referee Comment (RC2) · Anonymous Referee #2 · 15 Jan 2019

The manuscript presents an original approach in the evaluation of the so-called nesting technique used in regional climate modelling. The perfect model framework allows to perfectly isolate the detrimental impact of imposing lateral conditions to a limited area model. However the framework is somewhat different from the traditional RCMs: the forcing area is not a narrow band, but the rest of the globe. The paper is short and avoids non-essential descriptions, because some details might depend on the model and on the approach. This study deserves publication after a few clarifications and minor corrections listed below:

1. page 1 line 34: do not forget to mention here statistical downscaling

2. page 1 line 35: "impact" an impact model and an RCM are two different things

3. page 2 line 2: The climate of an RCM is generally better because of the higher horizontal resolution, but also because the empirical adjustments of the parameterizations are specific to its domain

4. page 2 line 4: "boundary", the relaxation area is not a boundary here, as the RCM has a global integration domain, and thus no boundaries

5. page 4 line 17: how do 90 min compare with the model time step and the frequency of updating the relaxation conditions ? The authors should mention that in an actual RCM the relaxation time generally varies between these two times from the inner to the outer relaxation zone.

6. page 5 line 38: what is the difference between "idealized" and "ideal" ? Do the authors oppose "simplified" to "accurate" ?

7. page 6 line 2 and further in the text: "autonomy" does not fit in the case of DS300-to-300 (it might be more suitable in the case of DS300-to-100). Indeed the day-by-day solutions of the RCM and GCM should be identical. The difference appears because of the numerical inadequacy of the driving, amplified by the non-linearity of the equations (similar to the butterfly effect in GCMs). If the forcing were perfectly adequate, the differences between the two models should be minimal, irrespective of the "autonomy" of the RCM.

8. page 7 lines 1-5: please discuss further this feature wrt lateral forcing (e.g. is T2M subject to horizontal advection ?) 9. page 7 line 20: "first ten"; this error is found further in several instances.

10. page 8 lines 29-34: the decreasing trend in the correlation is certainly significant. This is not necessarily the case of the fluctuations (e.g. EOF3 vs EOF4). To make the assessment clearer, the authors should calculate the correlation at each grid point. After a proper spatial filtering, they could observe a correlation minimum in the centre of the domain. Then, an EOF with its maximum weights in the centre is expected to

have a lower correlation.

11. page 9 line 16: reference ? (e.g. Michelangeli and Vautard)

12. page 12 line 3: stronger

13. page 12 line 11: comes

---

## Author Comment (AC2) · 22 Jan 2019

**Referee Comment**: The manuscript presents an original approach in the evaluation of the so-called nesting technique used in regional climate modelling. The perfect model framework allows to perfectly isolate the detrimental impact of imposing lateral conditions to a limited area model. However the framework is somewhat different from the traditional RCMs: the forcing area is not a narrow band, but the rest of the globe. The paper is short and avoids non-essential descriptions, because some details might depend on the model and on the approach. This study deserves publication after a few clarifications and minor corrections listed below:

**Authors Response**: Thank you for your constructive comments. We take into account

all your remarks in the new manuscript.

**Referee Comment**: 1. page 1 line 34: do not forget to mention here statistical down-scaling

**Authors Response**: It is mentioned now.

**Referee Comment**: 2. page 1 line 35: "impact" an impact model and an RCM are two different things

**Authors Response**: Yes. We agree. We removed the word "impact" to avoid confusion.

**Referee Comment**: 3. page 2 line 2: The climate of an RCM is generally better because of the higher horizontal resolution, but also because the empirical adjustments of the parameterizations are specific to its domain

**Authors Response**: Yes, we agree. We added a phrase in this sense.

**Referee Comment**: 4. page 2 line 4: "boundary", the relaxation area is not a boundary here, as the RCM has a global integration domain, and thus no boundaries

**Authors Response**: In our case, the Newtonian relaxation is applied only outside the RCM's domain. The buffer zone is indeed the whole globe outside the RCM's effective domain. We use the term "boundary" to respect the traditional concept for limited-area models.

**Referee Comment**: 5. page 4 line 17: how do 90 min compare with the model time step and the frequency of updating the relaxation conditions? The authors should mention that in an actual RCM the relaxation time generally varies between these two times from the inner to the outer relaxation zone.

**Authors Response**: The lateral boundary conditions from GCM are renewed every two hours. For the relaxation time scale controlling the relaxation strength, we used a binary solution: 90 minutes outside the domain and infinity inside (1.0e+25). We didn't

employ any transition between the two. In fact, our configuration inherited from a two-way nesting methodology in which the two models should be spatially complementary from each other and there is no recovering between them.

**Referee Comment**: 6. page 5 line 38: what is the difference between "idealized" and "ideal"? Do the authors oppose "simplified" to "accurate"?

**Authors Response**: We would like to maintain the use of "idealized" versus "ideal" (to make some humor for our writing), although we agree that an "idealized" protocol is a "simplified" one, but not necessarily an inaccurate one.

**Referee Comment**: 7. page 6 line 2 and further in the text: "autonomy" does not fit in the case of DS300-to-300 (it might be more suitable in the case of DS300-to-100). Indeed the day-by-day solutions of the RCM and GCM should be identical. The difference appears because of the numerical inadequacy of the driving, amplified by the non-linearity of the equations (similar to the butterfly effect in GCMs). If the forcing were perfectly adequate, the differences between the two models should be minimal, irrespective of the "autonomy" of the RCM.

**Authors Response**: We think the word "autonomy" is an appropriate description of RCM behaviors, although we agree completely that the butterfly effect is a fundamental cause to diverge the two models (RCM versus GCM). But there are two other explanations. First, the relaxation time is an e-folding time scale, relaxed variables can never reach the true relaxing values. Second, GCM provides driving conditions only every two hours, not every time step (half an hour in our model). These practices, together with the atmospheric nature of butterfly effect, ultimately give certain autonomy to RCM. In our idealized protocol of DS-300-to-300 (GCM and RCM identical, including spatial resolution), GCM and RCM would produce strictly identical results if the GCM updating frequency was the same as the model time step, and if the relaxation was replaced by a simple assignment (= in Fortran).

**Referee Comment**: 8. page 7 lines 1-5: please discuss further this feature wrt lateral

forcing (e.g. is T2M subject to horizontal advection?)

**Authors Response**: We recognize that we don't fully understand this behavior. We think that the RCM autonomy is certainly amplified by the interaction with the surface. This may explain why the divergence between GCM and RCM is larger for T2m than for Z500.

**Referee Comment**: 9. page 7 line 20: "first ten"; this error is found further in several instances.

**Authors Response**: That's done. Thanks.

**Referee Comment**: 10. page 8 lines 29-34: the decreasing trend in the correlation is certainly significant. This is not necessarily the case of the fluctuations (e.g. EOF3 vs EOF4). To make the assessment clearer, the authors should calculate the correlation at each grid point. After a proper spatial filtering, they could observe a correlation minimum in the center of the domain. Then, an EOF with its maximum weights in the centre is expected to have a lower correlation.

**Authors Response**: We put our largest effort here in re-calculating the correlation coefficients between GCM and RCM for each spatial grid. This was done for three emblematic variables: geopotential height at 500 hPa, 2-m surface air temperature and surface precipitation. Results are shown in Figure R2-1, together with RMSE on the right column. We can see that the intuition of Referee is partly true: larger correlation coefficients are generally found at borders and lower values inside. But it is also clear that there are many exceptions, the concept based on spatial structures, as what we want to emphasize in our manuscript, would be more appropriate. The intuition would be totally true if the atmosphere was motionless and the diffusion was the only process controlling the signal propagation in the atmosphere. The real atmosphere (hopefully also in a GCM) has plenty of dynamical motions: advection, convection and dynamical waves. They can easily break down the intuitive image.

[Figure]

**Referee Comment**: 11. page 9 line 16: reference? (e.g. Michelangeli and Vautard)

**Authors Response**: That's done. We added two references: Michelangeli and Vautard, 1995, and Vautard, 1990.

**Referee Comment**: 12. page 12 line 3: stronger

**Authors Response**: That's done.

**Referee Comment**: 13. page 12 line 11: comes **Authors Response**: That's done.

**Figure R2-1**: Correlation coefficients (left column) and Root-Mean-Squared Error (RMSE, right column) between GCM and RCM. The calculation was performed for the synoptic variability only and for the whole 80 years. Upper panels for geopotential height (m) at 500 hPa, middle panels precipitation (mm/s) and lower panels 2-m surface air temperature (°C).
* * *
[Figure]

**Fig. 1.** Figure R2-1

---

## Editor Comment (EC1) · Goelzer (Editor) · 23 Jan 2019

Dear authors,

As topical editor for this manuscript, I am posting this comment to inform you about the next steps in the revision process.

You will have seen that Anonymous Referee #1 has requested mayor revisions to clarify two main questions, one in relation to how Newtonian relaxation is applied in the model and the other about the nudging time scale. Given that comments of Anonymous Referee #2 refer to minor changes, I recommend that you upload a revised version, taking into account the comments of *both* referees. I will then ask Anonymous Referee #1 to re-evaluate the revised manuscript in perspective of the clarifications you have already

[Figure]

**[GMDD](/)**

Interactive
comment

posted in two author comments and anything else you want to add in form of a rebuttal.

Best greetings

Heiko Goelzer